# Development of a novel index to characterise arterial dynamics using ultrasound imaging

Joel Ward[1], Xinghao Cheng[2], Yingyi Xiao[2], Pierfrancesco Lapolla[1], Anirudh Chandrashekar[1], Ashok Handa[1], Robin A. Cleveland[2‡], Regent Lee[1‡]*

**1** Nuffield Department of Surgical Sciences, University of Oxford, John Radcliffe Hospital, Oxford, United Kingdom, **2** Institute of Biomedical Engineering, University of Oxford, Oxford, United Kingdom

☯ These authors contributed equally to this work.
‡ These authors also contributed equally to this work.
* regent.lee@nds.ox.ac.uk

**Data Availability Statement:** All relevant data are within the paper and its Supporting information files. The raw research data remains the property University of Oxford and can be made available to

## Abstract

Abdominal aortic aneurysms (AAA) are associated with systemic inflammation and endothelial dysfunction. We previously reported flow mediated dilatation (FMD) of the brachial artery as a predictor of AAA growth. We hence hypothesised that other physical characteristics of the brachial artery correlate with AAA growth. Using a prospectively cohort of AAA patients, we devised a 'brachial artery relaxation index' (BARI) and examined its role as a biomarker for AAA growth. However, no correlation between BARI and future aneurysm growth was observed (p = 0.45). Therefore, our investigations did not substantiate the hypothesis that other physical characteristics of the brachial artery predicts AAA growth.

## Introduction

Abdominal aortic aneurysms (AAA) are permanent and irreversible localised dilatations of the aorta involving all three layers of the vascular wall. The incidence of AAAs increase with age demonstrated through ultrasonography based screening studies diagnosing AAA in 1–2% of all 65–year-old men and in 0.5% of 70–year-old women [1, 2]. AAAs are usually asymptomatic so are most frequently detected either as part of screening programmes or incidentally on presentation to healthcare for an alternate pathology. AAAs have a tendency to expand, and as the size increases, so does the risk of rupture. When a AAA ruptures catastrophic intra-abdominal haemorrhage ensues which, if left untreated, results in mortality.

International guidelines recommend the A-P diameter threshold of 55mm for elective surgical repair in men, at which point the yearly risk of rupture outweighs the risks associated with an operation [3]. In addition to this, intervention is recommended for patients experiencing symptoms from their aneurysm, patients with saccular shaped aneurysms or those with aneurysms that expand quickly within one year [3]. Accordingly, patients with AAAs less than 55mm are typically kept under surveillance by regular ultrasound until the AAA expands above the threshold. Longitudinal studies have demonstrated that AAAs grow at different rates in different individuals and possess a variety of different physiological characteristics, the subtleties of which are not incorporated in current decision-making guidance [4, 5]. Although

other researchers based with material transfer agreement in place. This can be arranged through the Oxford University research services research.services@admin.ox.ac.uk).

**Funding:** The Oxford Abdominal Aortic Aneurysm Study was supported by the following: University of Oxford, Medical Sciences Division Medical Research Fund (MRF/HT2016/2191); University of Oxford, Nuffield Department of Surgical Sciences; John Fell Oxford University Press Research Fund (142/075); National Institute of Health Research (NIHR) Oxford Biomedical Research Centre; Regent Lee was supported by a Academy of Medical Science Starter Grant, UK (SGL013/1015). Pierfrancesco Lapolla was supported by an EU Erasmus+ traineeship studentship. Anirudh Chandrashekar is a Clarendon Keble Scholar, University of Oxford.

**Competing interests:** The patent relating to the material in the article has not altered our adherence to PLOS ONE policies on sharing data and materials. The patent filing number is PCT/GB2019/052161 and the title is 'Aneurysm Growth Rate Estimation'. There has been no consultancy, employment or product development related to this patent. This patent was filed based on the research work described in the manuscript and has not been commercialised. I confirm there is no conflict of interest to declare related to this.

70% of patients eventually meet the size threshold for an operation, they age and become more co-morbid during the observational period, raising the risks associated with intervention [4, 5]. Robust biomarkers of future AAA progression can improve clinical management by aiding identification of aneurysms which will progress to the necessitating intervention without requiring delay caused by a period of observation.

AAAs are associated with features of endothelial dysfunction throughout the cardiovascular system as well as a systemic inflammation response [6–8]. Endothelial function has been widely assessed using endothelial dependent function as a surrogate marker. One example of this includes flow mediated dilatation (FMD), which is a repeatable, non-invasive physiological assessment for the quantification of systemic endothelial function. FMD was shown to be inversely correlated with future AAA progression in humans; FMD deteriorates during the natural history of AAA, and is improved by surgery in a fashion predictable with modest accuracy [9].

We sought to improve the predictive power by incorporating further biomechanical data extractable from the ultrasound data loops already recorded for FMD to find another novel biomarker during the natural history of individuals with AAAs. We hypothesise that intrinsic physical viscoelastic characteristics of the brachial artery wall correlate with AAA growth.

## Material and methods

Details regarding the OxAAA study cohort and recruitment process have been published [9]. The Oxford Abdominal Aortic Aneurysm (OxAAA) study is designed to investigate longitudinally the natural history of AAAs. The study received full regulatory and ethics approval from the Oxford University and Oxford University Hospitals (OUH) National Health Service (NHS) Foundation Trust (Ethics Ref: 13/SC/0250). Every participant provided written consent. Participants were recruited at the John Radcliffe Hospital, which is part of the OUH NHS Foundation Trust.

The AAA size was defined by the maximal anteroposterior diameter (outer-to-outer). We defined further subgroups according to the AP diameter (APD) into small (30-39mm), moderate (40-55mm), and large (>55mm) AAAs. The annual growth rate of AAA during surveillance was calculated by: (ΔAPD/APD at baseline) / (number-of-days-lapsed/365days).

The clinical encounter can be a stressful experience for the study participants. To minimise the effect of this, we synchronised the research assessment with their regular NHS appointments for AAA surveillance scans, during daytime hours. This minimised the inconvenience to the participants. Participants were required to fast for 6 hours before the study (except their regular medications) and to refrain from caffeinated beverages and cigarette smoking for 24 hours prior to their visits. To minimise stress, the study was performed in a dimly lit, quiet room, with the patient in the supine position separate to their clinical appointment. This was described in our original publication detailing the methods of FMD measurement [9]. B mode ultrasound imaging of the brachial artery was then performed using the CX50 ultrasound machine (Phillips, Amsterdam, Netherlands) with a L12-3 probe which was mounted using a bespoke holder (Fig 1) to record brachial artery motion during four cardiac cycles. All data were anonymised before use in the work reported in this article.

### Image analysis

Ultrasound recordings were saved in DICOM format and imported into MATLAB® (The MathWorks Inc. Natick, Massachusetts, USA) for analysis. Brachial artery wall positions were detected using a modified version of a level set evolution (LSE) approach traditionally used in MRI information processing [10]. This LSE method exploits the difference in pixel intensity in the grey scale ultrasound to locate the contours. A user-defined rectangle was manually placed

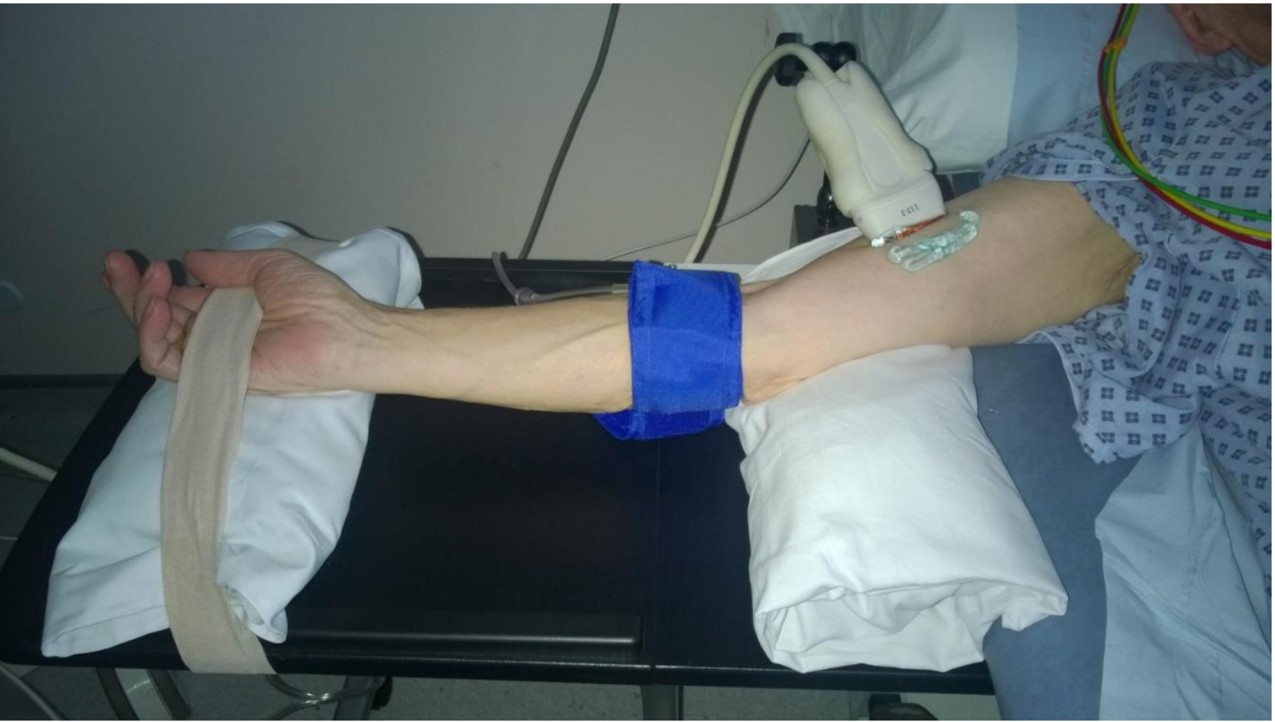

**Fig 1. Set up to record brachial artery ultrasound video loop.**

within the lumen position in the first frame of each ultrasound recording (Fig 2, panel A). The algorithm then iterated outwards towards the arterial wall detecting differences based on the brightness intensity contrast. The algorithm was designed to find the contours of the vessel outside the initial user-defined rectangle. Care was taken such that the rectangular region of interest is not close to lumen edges to prevent overlap with the arterial wall during cardiac cycles.

At each horizontal location in the image (x), the top of the lumen y$t$(x) and the bottom of the lumen $y_b$(x) were determined. This was repeated for each image in the cine loop which normally contained four cardiac cycles. Fig 3 shows an example surface plot of $y_B$(xH)ad $y_t$(x, t) where four cycles can be seen and the response is reasonably uniform along the length of the artery that was imaged. The diameter D as a function of x was determined by:

$$D = |y_b - y_t| \tag{1}$$

At each location x, the diameter D(x,t) was manually using the peaks and troughs so that systole and diastole could be identified, see Fig 4. The diameter of the vessel dilatation during systole was fit to a single exponential rise time.

$$\cup D_d(t) = A_d - B_d \times e^{-\left(\frac{t-C_d}{T_d}\right)} \tag{2}$$

Where $A$ and $B$, $C$ and $T$ are parameters to be fit. The relaxation during diastole was not well modelled by one time constant and therefore two time constants were employed to fit the diameter.

$$D_r(t) = A_r + B_{R1}e^{-\left(\frac{t-C_r}{T_{R1}}\right)} + B_{R2}e^{-\left(\frac{t-C_r}{T_{R2}}\right)} \tag{3}$$

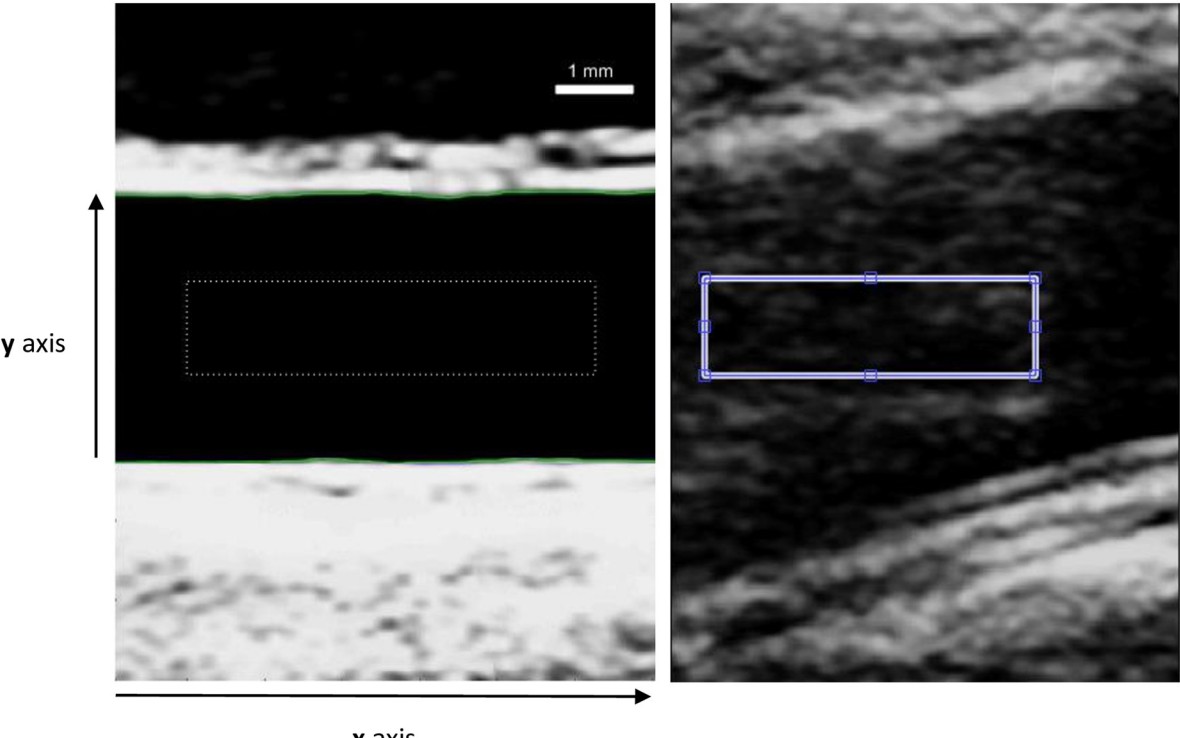

**Fig 2. Image of a frame extracted from the ultrasound video of a brachial artery.** The dotted rectangle is a user specified initialisation for the level set algorithm. **(A)** The green lines show the final estimate of the anterior and posterior vessel wall from the level set algorithm **(B)** Additional slice thickness artefact densities present in vessel lumen.

Here the relaxation time constants will be referred to as fast, $T_{R1}$ and slow, $T_{R2}$. The motivation for using exponential modes is that tissue is commonly modelled using as a series of springs and dashpots, the so-called Zener model which has exponential solutions [11]. For systole the response is a convolution of the driving pressure and vessel wall dynamics and so extracting wall properties is challenging. However, for diastole the temporal dynamics should be dominated by the properties of the wall which are captured by $T_{R1}$ and $T_{R2}$. The slow fall occurs under minimal arterial pressure and so is most sensitive to wall properties.

Assuming the incoming pulses are square waves with very short on duration, the tissue response is formed by three compartments in terms of the diameter: (i) steep rise (a convolution of the driving waveform and the tissue response), (ii) steep fall, (iii) slow fall before the next pulse comes. The steep rise and fall are caused by the forcing input, thus the composition of these two parts are mainly the input forcing term.

The best fit curves and time constants were constructed using the `fminsearch` function in MATLAB® (The MathWorks, Inc.), based on the Nelder-Mead simplex algorithm [12], in which convergence in low dimension has been verified in [13]. The cost function that is used in the minimalisation is the summation of standard L2-Norm, between the data and either Eqs 2 or 3.

The initialisation of parameters for `fminsearch` was as follows: The vertical (A_d, A_r) and horizontal (C_d, C_r) shifts' were set to the mean of the values' range within a fitting section to start with (diameter and time, respectively). The exponentials' magnitudes (B_d, B_r and D_r) were the size of one pixel. $T_{R2}$ was set to the time period of the relaxation section (ie diastole) and $T_{R1} = T_{R2} / 10$.

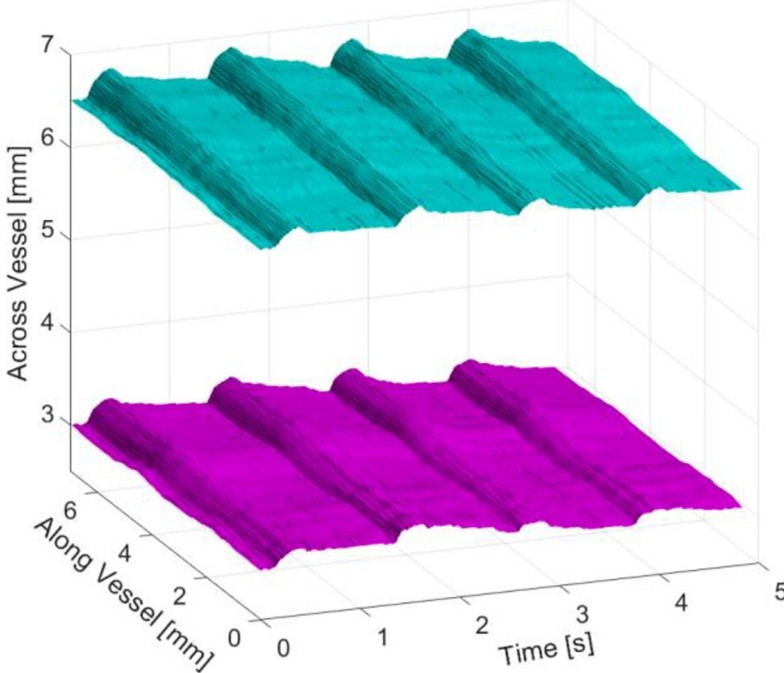

**Fig 3. Response of brachial vessel wall in time and space over four cardiac cycles.** The contour acquisition is conducted by the segmentation algorithm throughout the period of the ultrasound video clip. The expansion of the vessel during systole can be seen at about 0.5 s, 1.5 s, 3.0 s and 4.0 s. Cyan contours represent the posterior vessel wall. Magenta contours represent the anterior vessel wall.

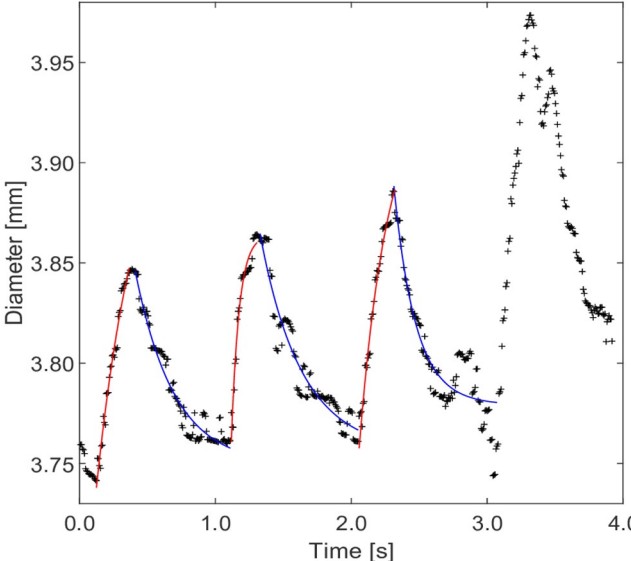

**Fig 4. Diameter variation as a function of time at a specific cross section showing four cardiac cycles.** The crosses show the measured results and the lines are the fitting curves for dilatation (red) and relaxation (blue). The diameter of the last cardiac cycle in the figure was not well fitted by a two time-scale decay and these data sets were excluded and not included in further analysis.

The brachial artery relaxation index (BARI) is defined by the mean relaxation constant of the exponential curves. The pilot data indicated that $T_{R2}$ was more likely to be a parameter differentiating aneurysm growth rate. Therefore, resulting analysis was focused on the diameter relaxation sections and the corresponding fittings.

## Results

Algorithm defined lumen wall positions were faithful to true brachial artery wall position in good quality ultrasound recordings (Fig 2, panel A) with an accuracy on the order of an image voxel (21 μm). The algorithm was unable to reliably locate wall position for 12 patients and these were excluded from further analysis. The most common issue encountered was the presence of a slice artefact in the lumen (Fig 2, panel B).

Matlab was used to create a 3D plot (Fig 3) displaying vessel wall movement, as a function of position along vessel and time through recording. As the segmentation algorithm decides how complicated the contour's shape must be to represent it at certain accuracy, the number of points representing each contour is generally different for a pair of top and bottom contours. In some cases, the segmentation returns "S" and "Z" shaped contours, means that for a single (x) position entry, there may be more than one contour (y) position values. The algorithm was written to include feasible manipulations to mitigate the above two issues in Eq 1. Before calculating the diameter, each contour is checked for whether there is more than one entry on a single horizontal position. If there is, the average is taken to be the value used in Eq 1.

For every location along the artery the diameter change was captured during cardiac cycles and curves fitted for dilatation and relaxation (Fig 4). The generated time constants were hypothesised to distinguish between fast and slow growing AAAs.

### Comparison 1: Pilot study

A pilot study was performed on six patients with the slowest (S1-3) and fastest (F1-3) growing aneurysms in the OxAAA cohort. The time constant for diameter dilatation sections ($T_D$) was not different between the two groups. BARI values are shown as bars in Fig 5 (green for slow growth AAA patients, red for fast growth AAA patients). The data suggests that BARI can be used to separate fast from slow based on a BARI threshold of 1.5s.

This positive result demonstrating separation in a pilot study for the patients at the extreme spectrum of prospective AAA growth and the plausible theoretical basis for the biomarker led to further analysis being performed on rest of the OxAAA cohort, comparisons and patient flow as demonstrated in Fig 6 (demographic data in S1 Table).

### Comparison 2: BARI between healthy volunteers and AAA patient cohort

16 healthy volunteers were age and gender matched to 16 patients from the aneurysmal cohort to assess whether BARI differed between patients with AAA (cohort) and without aneurysms (HV). There was no statistical difference observed between the two groups on unpaired t-test (HV median 0.93, HV interquartile range 0.65–0.99, Cohort median = 0.89, Cohort interquartile range = 0.82–1.03, P = 0.78).

### Comparison 3: BARI between AAA of different sizes

126 participants with a diagnosis of AAA were recruited to the study. BARI could be derived in 114 participants, as the video image quality were not sufficient for this calculation in 12 of them. Comparison is made of three subgroups according to their aneurysm AP diameter:

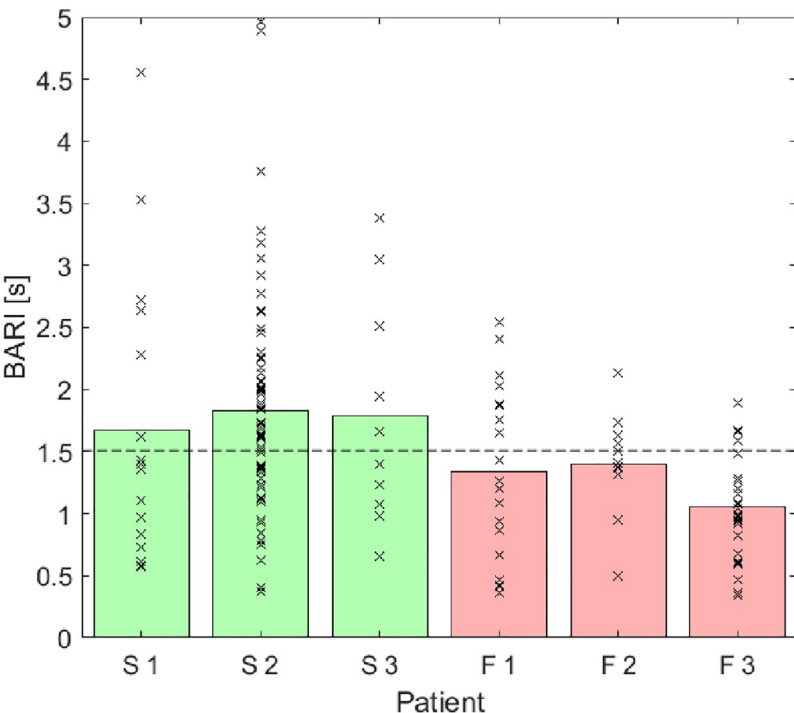

**Fig 5. Comparison of BARI between representative slow (S1, S2 and S3) growth and fast growth (F1, F2 and F3).**
Relaxation time constant values are marked in crosses for each patient. The bars represent the BARI values derived
from the data crosses for each patient (Green: slow growth patients. Red: fast growth patients). Dashed line at
BARI = 1.5 suggests a threshold value to separate patient's growth type.

small (30-39mm), moderate (40-55mm), and large (>55mm). There was no significant difference observed between groups on Kruskal-Wallis (Small vs Large, P = 0.82, Small vs Medium, P = 0.99, Medium vs Large, P = 0.99). Spearman correlation of BARI against aneurysm size (mm) was not significant (p = 0.11), Fig 7.

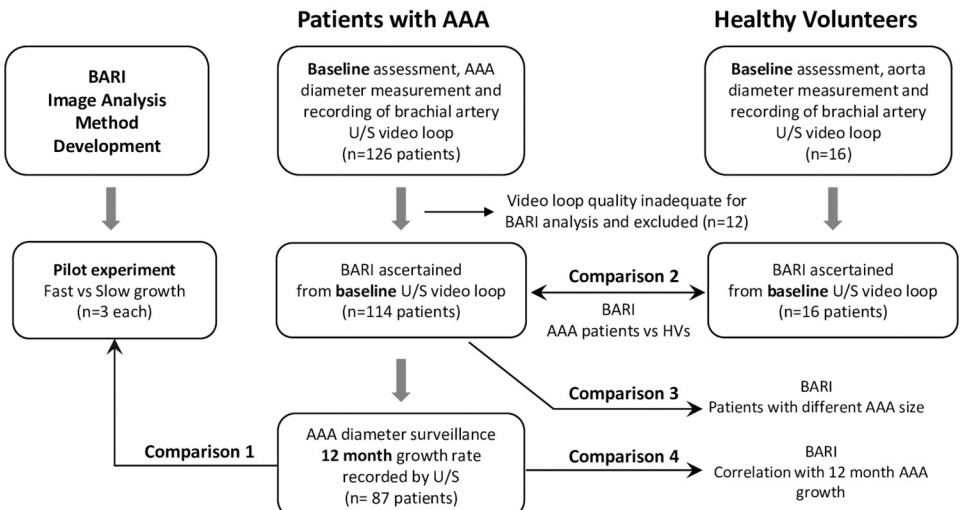

**Fig 6. Patient journey through OxAAA study.** Depicts data collection points and comparisons performed in this
workflow.

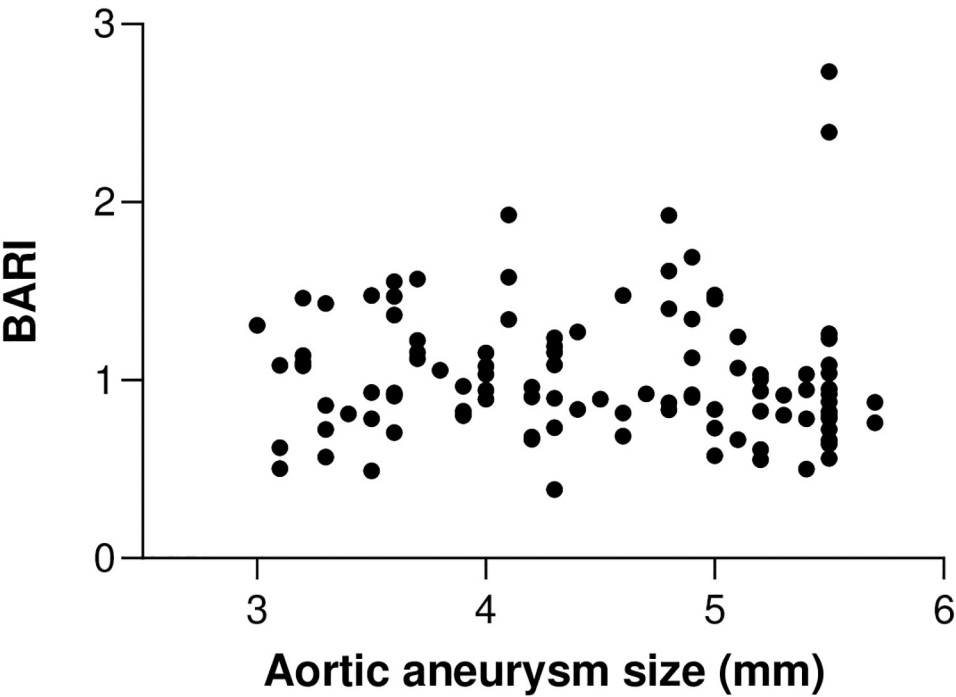

**Fig 7. BARI vs AAA size for 114 patients.** No correlation between aneurysm AP diameter (x-axis) and BARI (y-axis) P = 0.68.

## Comparison 4: Correlation between BARI and future AAA growth

The growth rate of individual participants was calculated over 12 months in 87 participants with complete follow up during this period. Fig 8 shows measured BARI for each patient versus the AAA growth and no correlation was observed. Spearman correlation of BARI against % aneurysm growth per year was not significant (p = 0.45).

## Discussion

This work was motivated by the desire to predict the growth and progression of AAAs. The existing surgical threshold based on a size criteria (>55mm) has intrinsic shortcomings, as aneurysm size is not an absolute predictor of the risk of rupture, and the rate of AAA progression varies greatly between individuals. Those with moderate size aneurysms have a small (1%) annual risk of rupture and the majority of patients (70%) with an initially small or moderate size AAA will progress and require surgery within 5 years [4, 5]. Improved biomarkers of future AAA progression would enable the stratification of patients who may benefit from surgery earlier than the existing size threshold and decrease the perioperative risk to those patients.

Our hypothesis was that US based measurements of the deformation of vessels walls could be used to extract mechanical properties of the vessels to provide information that could improve the ability to predict AAA progression. Preliminary data suggested that BARI, a measure of a relaxation time scale had the most promise. Unfortunately, there was no significant difference in BARI between aneurysmal patients and healthy volunteers, nor did it vary according to aneurysm size. It offered no predictive value for aneurysm growth at 12 months. Although BARI did not appear to improve the ability to predict growth in AAAs, the

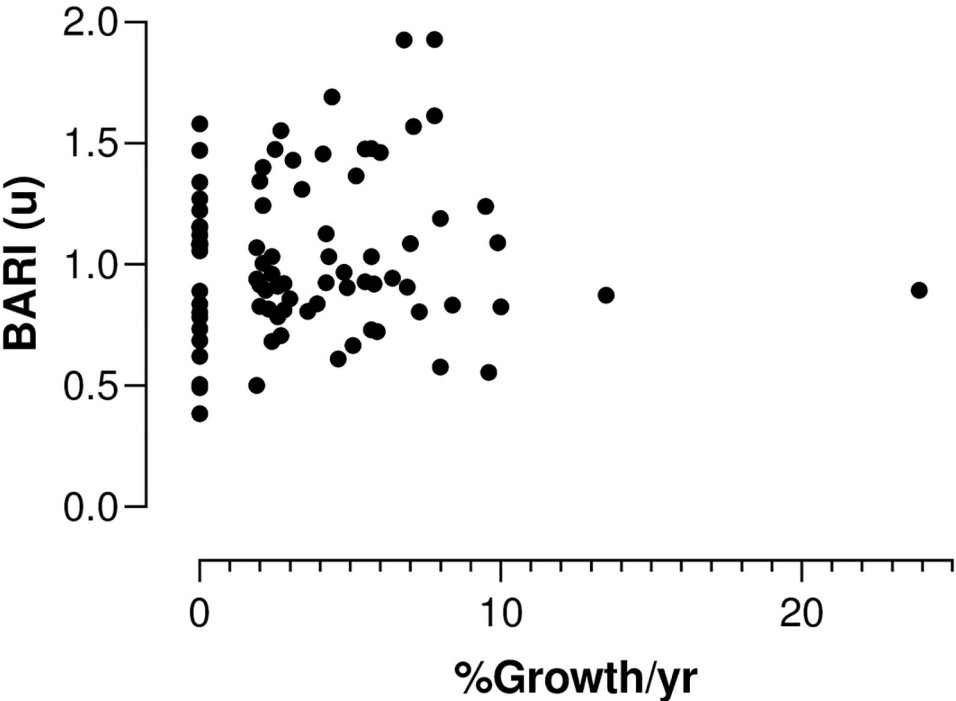

**Fig 8. BARI vs AAA % growth per year for 87 patients.** Linear regression comparing aneurysm % AP diameter growth per year (x-axis) and BARI (y-axis) Slope 0.01 P = 0.46.

processing approach described here to extract mechanical properties from ultrasound image analysis may have alternate applications in other pathologies or the research setting.

Investigating physiological characteristics of vasculature using imaging is an important part of developing our understanding of cardiovascular and aneurysmal disease. Better definition of the biophysical characteristics of vasculature in healthy and pathological states is important to help underpin future research. Ultrasound is an ideal imaging modality to extract bio-mechanical properties as it is non-invasive, functional, widely accessible and without exposure to ionising radiation.

As we did not observe correlation between BARI and future AAA growth, we did not proceed to examine potential confounding effects of the other demographic variables. As a post hoc analysis, we examined the association between BARI and the demographic variables as summarised in S1 Table. It is interesting that BARI had significant correlation with some demographic variables (gender, history of smoking, prior diagnosis of hypertension or diabetes). We further observed a significant correlation between BARI and aneurysm growth in female, however the sample size of females was too small. That BARI does not correlate with mean arterial pressure or HbA1C indicate that these patients have received appropriate medical therapy for their prior diagnosis of hypertension and diabetes.

We had hypothesised BARI would help the predictive index for aneurysm growth, in addition to existing biomarkers such as FMD, AAA diameter and circulating plasma proteins [9, 14]. Our data did not support this. It would have been preferable to perform the same analysis using ultrasound imaging of the AAA itself. However, as the AAA is an organ structure deep inside the abdominal cavity, ultrasound imaging of the AAA is only feasible using a low frequency probe (typically at 3MHz) to achieve the depth penetration. The images acquired using this probe were found to be too low in resolution for the analysis described here to extract

pulsatile expansion of the AAA. Nevertheless, BARI is a novel image analysis technique which underscores the importance of active pursuit of imaging biomarkers, across various modalities to help inform clinical decision making for AAAs.

## Conclusion

We developed a novel method to characterise the viscoelastic properties of brachial artery (BARI). Although BARI did not correlate to future AAA growth as hypothesised, our work-flow demonstrates a pipeline of interdisciplinary research which can be applied to other areas of research.

## Supporting information

**S1 Table. Summary of participant demographics and their association with BARI.** Partici-pant baseline demographics were collected at the time of enrollment to the study. Characteris-tics that follow a Gaussian/Normal Distribution are presented using mean ± SD, and cohort differences are compared using a student t-test. For variables that don't follow a Gaussian dis-tribution, median and inter-quartile range (IQR) are presented and cohort differences are compared using a Mann-Whitney test. BMI: body mass index. CAD: coronary artery disease, PAD: peripheral arterial occlusive disease, ACEi: angiotensin converting enzyme inhibitor; ARB: angiotensin receptor blocker.
(PDF)

**S1 Data. Spreadsheet of anonymised dataset as analysed.** Gender: 1—Male, 0 –Female BMI: body mass index, MAP: mean arterial pressure, HTN: hypertension ACEi: angiotensin con-verting enzyme inhibitor; ARB: angiotensin receptor blocker.
(XLSX)

## Acknowledgments

Contributors to the OxAAA Study include: Amy Jones, Felicity Woodgate, Nicholas Killough, Kirthi Bellamkonda, Sowmya Mangipudi, Members of the Oxford Regional Vascular Services and the Jackie Walton Vascular Studies Unit.

## Author Contributions

**Conceptualization:** Xinghao Cheng, Ashok Handa, Robin A. Cleveland, Regent Lee.

**Data curation:** Joel Ward, Yingyi Xiao, Pierfrancesco Lapolla, Anirudh Chandrashekar.

**Formal analysis:** Joel Ward, Xinghao Cheng, Yingyi Xiao, Pierfrancesco Lapolla, Anirudh Chandrashekar.

**Funding acquisition:** Ashok Handa, Robin A. Cleveland, Regent Lee.

**Investigation:** Joel Ward, Ashok Handa, Robin A. Cleveland.

**Methodology:** Joel Ward, Xinghao Cheng, Yingyi Xiao, Robin A. Cleveland, Regent Lee.

**Resources:** Pierfrancesco Lapolla, Ashok Handa, Regent Lee.

**Software:** Xinghao Cheng, Anirudh Chandrashekar, Robin A. Cleveland.

**Supervision:** Ashok Handa, Robin A. Cleveland, Regent Lee.

**Validation:** Regent Lee.

**Visualization:** Regent Lee.

**Writing – original draft:** Joel Ward, Xinghao Cheng.

**Writing – review & editing:** Robin A. Cleveland, Regent Lee.

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
