## [Decision Letter · Decision Letter 0]

26 Jan 2021

PONE-D-20-40123

Development of a Novel Index to Characterise Arterial Dynamics Using Ultrasound Imaging

PLOS ONE

Dear Dr. Lee,

Thank you for submitting your manuscript to PLOS ONE. After careful consideration, we feel that it has merit but does not fully meet PLOS ONE’s publication criteria as it currently stands. Therefore, we invite you to submit a revised version of the manuscript that addresses the points raised during the review process.

We look forward to receiving your revised manuscript.

Kind regards,

Athanasios Saratzis

Academic Editor

PLOS ONE

Additional Editor Comments:

Thank you very much for submitting this interesting Article to PLOS ONE.

Some major comments / issues were raised by the Reviewers, who are both world-class experts in diagnosis and management of aneurysmal disease. We would be most grateful if you could please address all points raised, but most importantly explain why this specific artery was chosen (radial). Please address the comment(s) raised about other arterial segments such as the popliteal artery (especially since this is the most common site of aneurysmal disease).

The manuscript will have to be re-evaluated by the Reviewers given their feedback.

2.) We note that you have a patent relating to material pertinent to this article. Please provide an amended statement of Competing Interests to declare this patent (with details including name and number), along with any other relevant declarations relating to employment, consultancy, patents, products in development or modified products etc. Please confirm that this does not alter your adherence to all PLOS ONE policies on sharing data and materials, as detailed online in our guide for authors http://journals.plos.org/plosone/s/competing-interests by including the following statement: "This does not alter our adherence to  PLOS ONE policies on sharing data and materials.” If there are restrictions on sharing of data and/or materials, please state these. Please note that we cannot proceed with consideration of your article until this information has been declared.

3.) We note that you have indicated that data from this study are available upon request. PLOS only allows data to be available upon request if there are legal or ethical restrictions on sharing data publicly. For information on unacceptable data access restrictions, please see http://journals.plos.org/plosone/s/data-availability#loc-unacceptable-data-access-restrictions.

4.) Please include captions for your Supporting Information files at the end of your manuscript, and update any in-text citations to match accordingly. Please see our Supporting Information guidelines for more information: http://journals.plos.org/plosone/s/supporting-information.

5.) We noticed you have some minor occurrence of overlapping text with the following previous publication(s), which needs to be addressed:

- Flow Mediated Dilatation and Progression of Abdominal Aortic Aneurysms (https://doi.org/10.1016/j.ejvs.2017.03.001).

In your revision ensure you cite all your sources (including your own works), and quote or rephrase any duplicated text outside the methods section. Further consideration is dependent on these concerns being addressed.

Reviewers' comments:

Reviewer's Responses to Questions

**Comments to the Author**

1. Is the manuscript technically sound, and do the data support the conclusions?

Reviewer #1: Yes

Reviewer #2: Yes

2. Has the statistical analysis been performed appropriately and rigorously? 

Reviewer #1: Yes

Reviewer #2: Yes

3. Have the authors made all data underlying the findings in their manuscript fully available?

Reviewer #1: Yes

Reviewer #2: Yes

4. Is the manuscript presented in an intelligible fashion and written in standard English?

Reviewer #1: Yes

Reviewer #2: Yes

5. Review Comments to the Author

Reviewer #1: An interesting paper. I agree that other biomarkers would be useful to predict AAA growth and that size is too simplistic.

Can the authors comment on the following-

1. Why choose the brachial artery? Ease of access is obvious but there is some data to suggest that AAA is a systemic dilating disease and the popliteal artery (with its association with AAA) was chosen

2. Have you allowed for confounding factors? Many of these patients will also have other risk factors (high BP, smoking etc) - does this affect your results? Do drugs affect the relaxation index? The patients were all studied during a clinical visit that can be stressful? Did you make allowances for this?

3. A null hypothesis and power calculation would help

Reviewer #2: Thank you for your well-written paper. I have the following comments:

1. Page 3, Lines 22: The threshold of 55mm is not the only criterion. The expansion rate within one year as well as the shape of the aneurysm sac are playing an important role as well.

2. Page 4, Line 39: Which is exactly the predictive power in percentage of the FMD in the brachial artery regarding the AAA identification?

3. Is the BARI and index that it has been evaluated in other studies?

4. The different number of patients presented in the results is very confusing. It would be meaningful to provide a flow diagram with all measurements performed and the respective cohorts of patients.

5. Page 10, Line 193: Please provide literature about these preliminary data.

6. Page 10, Line 195: How time-consuming is to measure the BARI and how can we simplify it in order to investigate further cardiovascular pathologies

7. Why did you select brachial artery and not radial artery? Is the size that makes the difference? Why not CFA or carotid artery or even better popliteal artery which is the second more frequent artery with aneurysmatic degeneration?

8. Do you think that further improvement of US device could provide more data regarding direct measurement of BARI in the abdominal aorta?

6. PLOS authors have the option to publish the peer review history of their article (what does this mean?). If published, this will include your full peer review and any attached files.

Reviewer #1: **Yes: **Robert Sayers

Reviewer #2: No

---

## [Author Response · Author response to Decision Letter 0]

12 Feb 2021

Please refer to the attached Response to Reviewer document for detailed response to reviewer feedback including new figures and references.

---

## [Editor Report · Decision Letter 1]

19 Feb 2021

Development of a Novel Index to Characterise Arterial Dynamics Using Ultrasound Imaging

PONE-D-20-40123R1

Dear Dr. Lee,

We’re pleased to inform you that your manuscript has been judged scientifically suitable for publication and will be formally accepted for publication once it meets all outstanding technical requirements.

Kind regards,

Athanasios Saratzis

Academic Editor

PLOS ONE
---

## [Editor Report · Acceptance letter]

23 Feb 2021

PONE-D-20-40123R1 

Development of a Novel Index to Characterise Arterial Dynamics Using Ultrasound Imaging 

Dear Dr. Lee:

I'm pleased to inform you that your manuscript has been deemed suitable for publication in PLOS ONE. Congratulations! Your manuscript is now with our production department. 

Kind regards, 

on behalf of

Dr. Athanasios Saratzis 

Academic Editor

PLOS ONE